

# Hydrodynamic characterization of past flash-flood events and their associated hazards from dendrogeomorphological evidence in Caldera de Taburiente National Park (Canary Islands, Spain)

Julio Garrote[1]; Andrés Díez-Herrero[2]; José M. Bodoque[3]; María A. Perucha[2]; Pablo Mayer[4]; Mar Génova[5]

[1] Department of Geodynamics, Complutense University of Madrid, Madrid, E-28040, Spain
[2] Geological Survey of Spain (IGME), Ríos Rosas 23, Madrid E-28003, Spain
[3] Mining and Geological Engineering Department, University of Castilla-La Mancha, Toledo, E-45071, Spain
[4] Department of Geography, University of Las Palmas de Gran Canaria, Las Palmas de Gran Canaria, E-35003, Spain
[5] Forest Engineering Technic School, Technic University of Madrid, Madrid, E-28040 Spain

*Correspondence to*: Julio Garrote (juliog@ucm.es)

**Abstract.** Las Angustias river is an ungauged stream located in the Caldera de Taburiente National Park (La Palma, Canary Islands, Spain), where frequent intense flash-flood events occur, sometimes with fatal consequences (4 deaths, November 2001) and considerable financial implications (over 700 000 euros in recent years). The aim of this research is to analyse the flood hazard at the Playa de Taburiente, one of the most popular sites in this protected area, with more than 60,000 visitors per year.

The use of classic data sources and hydrological or hydraulic modelling methods for flood hazard analysis has important limitations in this area because of incomplete precipitation and flow data information and low representative values of the statistical time series, which may lead to under- or over-estimated results.

Alternative or complementary data sources and methods including palaeohydrological techniques can therefore be used here for flood hazard analysis. A detailed dendrogeomorphological study of the river system was carried out using Canarian pine trees located on the stream bed and river bank with external evidence of flash-flood damage, including scars and exposed roots.

The preliminary results identify and date at least eight winter flood events between 1962-1963 and 2011-12. In spite of the uncertainties deriving from the incomplete precipitation data and the mobile alluvial riverbed, the models provide an estimate of past flood discharge magnitudes. E.g. for the 1997 flood event a 1235 $m^3 \cdot s^{-1}$ flood minimizes the RMSE over the disturbed tree sample; furthermore, this flow value clearly exceeds the return period considered and means a distinct behavioral change in this gorge, from a braided channel with emerged bars to a single channel occupying the whole river bed. These numerical results and maps could improve flood hazard and risk analysis and should be useful for the national park land use management and visitors planning.



# 1 Introduction

Flash floods are common in mountain basins, as a result of their high slopes and quasi-circular morphology, the resulting strong connectivity and high order streams (Nakamura et al., 2000; Viviroli et al., 2003; Rickenmann and Koschni, 2010). The precipitation in these basins also has an important orographic component and as a result is very variable in spatio-
temporal terms (Roe, 2005; Rotunno and Houze, 2007). This means that mountain basins are highly prone to extreme precipitation events, both in total volume and intensity. The resulting floods have a rapid hydrological response, characteristically defining peaky hydrographs (i.e. short lag time), reaching their peak flows within a few hours, and thus allowing little or no advance warning to predict flood damage (Borga et al., 2007; Borga et al., 2008). These basins usually have high landscape value and recreational use (Seger, 2009) and with the characteristic hydrological response of mountain
basins this poses a considerable risk to life, as was evidenced in the 1996 Biescas (Spain) disaster (White et al., 1997; Benito et al., 1998).

Flash-floods in the Caldera de Taburiente National Park have caused the death of several hikers in an area with over 65 000 visitors using the hiking trails each year. Four people died in 2001 when they were swept away while walking along the bottom of the ravine. Many people have been stranded in the park as a result of rising river levels during floods, leading to
rescue operations and temporary closure of the trails. In October 2011 more than forty visitors were stranded in the camping area, and had to be rescued by helicopter. Finally, flash-floods have considerable financial implications for the national park infrastructure with damage to trails, bridges and aqueducts and for reforestation projects using endemic species which are currently underway. Since 2009 the managers of the national park have spent considerable resources (over 700,000 euros just in 2010) on planting endemic willow species (Salix canariensis) in the alluvial river bed. During the first flood event
after the plantation most of the established endemic trees were carried out to the Atlantic Ocean. Flood risk analysis and planning is therefore essential for national park managers, civil protection and other authorities (water resources managers).

Flood risk analysis and management is clearly conditioned by data availability. Flow quantile estimates can be obtained directly by statistical analysis (FFA; flood frequency analysis) of maximum discharge provided that the time series are statistically significant. When flow data are unrepresentative but there are enough precipitation data the alternative is to run,
calibrate and validate hydrologic rainfall-runoff models, which provide output data to determine the flooding area and the hazard parameters using hydraulic models. However, there is often insufficient data in mountain basins, either because the data simply do not exist or the accuracy is poor in terms of spatio-temporal statistical significance. As a result, indirect flood characterization methods are increasingly being used as an alternative (Gaume and Borga, 2008). In this context, different palaeostage indicators (PSI) and high water marks (HWM) have been used to characterize past floods, forming the basis of
palaeohydrology (Benito and Thorndycraft, 2005) and post-event flood analysis (Gaume, 2006). Indirect data sources can be grouped as: i) geological/geomorphological; ii) historical; and iii) botanical.

Geological and geomorphological data sources include sediment deposits and erosion lines or marks. These can be used to relate this evidence to flow characteristics, thus enabling reconstruction of flow and other physical flood parameters using



empirical formulas or mathematical models (Benito, 2002). Historical methods are based on the documentation available in historical or newspaper archives and libraries. Permanent high-water flood level markers on buildings, as well as oral and audiovisual testimonies can be used as additional data sources. These enable the reconstruction of the flooded area and therefore of the flood depths reached by integrating historic data into hydraulic models. The levels inferred from this data

can then be transformed into flows and integrated into the systematic records, reducing frequency analysis uncertainty (Benito et al., 2004; Brazdil et al., 2006). Finally, evidence of flood occurrence can also be recorded in the type and structure of riparian vegetation, using a dendrogeomorphological approach (Diez-Herrero et al., 2013). This method obtains improved frequency analysis using a similar approach to that used with historical data. It also reduces uncertainty in hazard and risk analysis where it enables calibration of roughness and other sensitive parameters (Ballesteros et al., 2011).

Apart from the methodological aspects, the aim of this study is the hydrological and hydraulic characterization of river past flash-floods in ungauged mountain basins using dendrogeomorphological evidence, and conditioned by a movable river bed environment. To achieve this aim it will focus on an analysis of the flood event which occurred 11- 13 January 1997 in the Caldera de Taburiente National Park. Using this extreme river event as a case study we will attempt to: draw conclusions as to the water and sediment load flows in the basin on these dates; establish methodological features, highlighting the strengths

of this type of analysis; and finally show the limitations and uncertainty of the results obtained, that can be used for future hazard and risk analysis of floods in this area.

## 2 Case Study Area

The Caldera de Taburiente National Park is on the island of La Palma, the north-westernmost island of the Canary Islands archipelago, Spain (Figure 1). Declared a national park in 1954, this area of 47 km2 is situated centrally in the northern part

of the island (28°40'- 28°46' N, 14°08' - 14°13' W (Greenwich Meridian)). The altitude ranges from 2426 m a.s.l. (Roque de Los Muchachos) to 430 m a.s.l. in the Barranco de Las Angustias just on the edge of the park.

The main part of the park is formed by the headwaters of the Barranco de Las Angustias, a semi-circular basin 8 km in diameter and 2000 m from top to bottom, resembling a great volcanic crater or caldera, although its morphology is in fact the result of the superposition of several volcanic edifices, erosive phases and large landslides (Colmenero et al., 2012). This

research focuses on the central area of the park, known as 'Playa de Taburiente', a Y-shaped 2 km long stream reach, formed by two gorges, the 'Verduras de Alfonso' and 'Cantos de Turugumay' ephemeral rivers, and the Taburiente River, the result of their confluence (Figure 1).

### 2.1 Geology

The Canary Islands were formed by volcanic activity around 20 Ma ago with an approximately westward age progression,

with La Palma and El Hierro the two most recent islands of the archipelago. The volcanic history of La Palma originated around 4 Ma ago (Staudigel et al., 1986) when a seamount formed on the Jurassic oceanic crust and was subsequently



elevated to an altitude of up to 1500 m a.s.l., where it now forms the Basal Complex, made up of basaltic and trachytic pillow lavas, hyaloclastites and breccias, mafic plutonic bodies and dike swarms of alkaline basaltic composition. Subaerial activity started around 2.0 Ma (Ancochea et al., 1994), when a multi-stage shield volcano was formed. This central edifice experienced at least one major collapse resulting in a landslide and subsequent rebuilding of the edifice (Ancochea et al., 1994; Carracedo et al., 1999).The subaerial volcanic reactivation, with explosive volcanism predominating during its initial stages producing abundant volcanoclastic and phreatomagmatic materials at the base of the subaerial edifice, continued until at least 0.41 Ma. The landslides on the Taburiente and Cumbre Nueva around 0.7 Ma (Ancochea et al., 1994) resulted in the formation of the Caldera de Taburiente by incision and retrogressive erosion.

## 2.2 Precipitation

The three main characteristics of the rainfall in the Caldera de Taburiente are: (i) interannual irregularity; (ii) seasonality; (iii) torrentiality. These characteristics are also evident on the E and SE slopes of other islands in the Canary archipelago (Marzol et al., 2006; Dorta, 2007). The mean annual rainfall is 1018.6 mm but this is irregularly distributed (i), so the coefficient of variance is 51.7% and the standard deviation 526.8. The marked seasonality (ii) is evident as over half of all precipitation falls during the winter but in the summer months when the islands are affected by the Azores anticyclone, less than 1% of the annual rainfall is recorded. Finally, the torrential rainfall (iii) is one of the most significant traits as in some months several times the average monthly rainfall has been recorded; as e.g. more than seven times in January 1979; or six times in February 2010. The torrential nature of the precipitation is also evidenced by the 200 mm of rainfall in a single day recorded on 11 occasions in just 22 years; this amount of rain therefore represents a significant percentage of the monthly and annual totals (average 40% and 25% respectively).

## 2.3 Vegetation and land use

The Pinus canariensis forest is the most characteristic vegetation of the Caldera de Taburiente where it forms extensive forests and nowadays is one of the best preserved forests of this species in the Canary Islands. On the riverbed there are occasional groves of willows (Salix canariensis) and some stands of the less demanding laurisilva species in the most protected shaded areas; while at higher altitudes the plant cover is formed by different shrub species (Arco et al., 2006). Although the Caldera de Taburiente is very sparsely populated due to its rugged topography, the research results of Garzón-Machado et al. (2010) indicate that the Canary pine forest has been severely impoverished by herbivore activity, probably resulting from the introduction of goats by the auaritas, the original inhabitants of La Palma. Other land uses include forest harvesting, hunting and a few relatively recent rainfed crops of low economic value which share the land with grazing.



## 3 Material and Methods

The general approach used in the study methodology is summarized in the diagram in Figure 2, showing the analytical study combining various data sources (dendrogeomorphological, meteorological, topographical) and methods (dendrochronological, meteorological, hydrological, hydraulic and geomorphological) which through successive approaches characterize recent flood events in the latter half of the 20th century and their potential hazard and risk.

### 3.1 Dendrogeomorphological data sources

63 wounds of 54 trees were dated in the Caldera de Taburiente corresponding to eight different years between 1962 and 2012 (Génova et al., 2015). The injuries occurring in 1962 (affecting almost all the older trees and with the longest scars measured) and in 1997, were especially relevant, with both presenting a large number of replications (48% and 22%, respectively, of the total trees; Figure 3). All sampled tree data were recorded in a spatial database and scar sizes were also logged, to relate the year of injury, geographical position of each tree, wound / scar size and elevation. Of all the injuries dated, the event associated with 1997 was chosen for the case study, because of the available precipitation information and number of samples with valid disturbances (11).

### 3.2 Hydrological analysis

The unique flow data available could come from the gauging station located in the Dos Aguas river reach (mid Las Angustias gorge), used for water resources derivation and irrigation. However, this weir dyke only measures discharges during ordinary flow, not during flood events, which drain directly to the Atlantic Ocean. There is therefore no flow data available for past flood events, and thus it is impossible to apply flood frequency analysis (FFA) using discharges. As a result, the hydrological flood analysis was carried out using rainfall-runoff models.

### 3.2.1 Precipitation analysis

Data was used from two stations, a pluviometer (daily data), located inside the Caldera de Taburiente National Park, and a continuous record pluviograph (hourly data) installed in La Palma airport, thirty kilometres far from the basin (Figure 1). Other measuring stations are installed in the Caldera, including another rain gauge situated at a lower altitude than the area of interest with a short, poor quality record series; there are also other gauges (monthly total data) installed at higher altitudes, but their monthly values were not considered useful for the aims of this case study.

The pluviograph enabled the definition of the shape and duration of the hyetographs associated with intense rainfall, and these were then used to generate synthetic hyetographs (symmetrical, right-skewed and left-skewed distribution) associated with rainfall events in the years where the dendrogeomorphological data show disturbances.

Additionally, using data obtained from the rain gauge installed in the Caldera, a frequency distribution analysis was carried out to estimate the precipitation quantiles associated with different return periods. These quantile estimates (Coles, 2001)





were obtained from annual maximum series (SQRT-ETMAX and GEV functions), and from exceedance series (Generalized Pareto – POT).

The use of exceedance series to estimate or assign frequencies to extreme events is supported by many previous studies (e.g. Wang, 1991; Coles et al., 2003; Wilks, 2010), and has been applied to both flow and precipitation estimates.

These exceedance series were also derived from two types of precipitation events occurring in the study area: events with a duration of under 24 hours, defined by an absence of precipitation recorded in the 24 hours previous to and following this rainfall event; events with a duration of over 24 hours also defined by the absence of precipitation during the 24 hour period immediately before and after the event.

In both cases, the precipitation threshold (the process was carried out by the graphic method) value adopted was similar at 25
mm rainfall. This value produced an exceedance series of 357 values above the threshold for precipitations of less than 24 hours and 185 values for those lasting more than 24 hours, both values significantly higher than the length of the annual maximum series (37 years).

Based on the above, the frequency function chosen was the Generalized Pareto, with parameters of fit estimated by the L-Moments method; applied to the exceedance series over the threshold.

The event analysed lasted from 11-13 January 1997. The duration was therefore over 24 hours and the information provided for this event by the rain gauge installed in the national park allows a justified distribution of the precipitation volume for each of the quantiles obtained from the analysis of the exceedance series above the threshold. The time distribution of the precipitation throughout the event is important, as this will have a clear influence on parameters such as initial abstractions and runoff generation and therefore on the flows generated. These data were used to generate synthetic hyetographs using
the alternating blocks method (Chow et al., 1994), for the three day duration of the event.

### 3.2.2 Rainfall -runoff modelling

 The basin model (Figure 4) was derived from a 5 m Digital Elevation Model (DEM). Disaggregation was based on both the spatial distribution of the physiographic factors that determine a homogenous hydrologic response (i.e., lithology, soils, cover type, hydrologic conditions and slope) and those control sections where dendrogeomorphic data was available. The
curve number method (USDA-SCS, 1972; Boughton, 1989) was applied to determine initial abstractions, based on land use, land cover type and hydrologic soil group.

The runoff hydrograph was obtained using the Clark unit hydrograph (Adib et al., 2011), and time of concentration (tc) and storage coefficients had to be estimated. Time of concentration was derived using the Témez formula (Rico et al., 2001) which depends on the length of the main channel for each sub-basin and its slope. Storage coefficient was calculated
assuming it represents 0.6 tc. The Muskingum-Cunge method (Ponce and Yevjevich, 1978) was implemented for the routing of the flood wave, due to the absence of observed flow data. The Muskingum-Cunge parameters were calculated based on the flow and channel characteristics (Ponce, cited by Tewolde, 2005). This method involved the use of a finite difference scheme for solving the Muskingum equation, where the parameters in the Muskingum equation were determined based on





the grid spacing for the finite difference scheme and the channel geometry characteristics. The geometry and roughness of the channels were accurately characterized using LIDAR (0.5 points per m2) data combined with GIS tools.

## 3.3 Hydraulic modelling

The hydraulic modelling was carried out with numerical simulation using a finite differences model (St Venant 2D equations), using the Iber hydraulic simulation free software (Bladé et al., 2014), with the following characteristics:

The boundary conditions related to inflows were restricted to the peak flow since the hydrographs obtained from the rainfall-runoff modelling present high uncertainty and reduce reliability using the complete hydrograph. Thus two flow groups were modelled:

- Peak flows associated with precipitation recorded during 11-13 January 1997 event, obtained from the hydro-meteorological model.

- Peak flow range (300 - 2500 $m^3 \cdot s^{-1}$), to minimize errors (RMSE) compared with the vegetation disturbance data (tree scars), by comparing the water sheet depths with the height of the disturbances. The aim here was to define the liquid flows needed to minimize this error.

The topographical information relating to the channel geometry of the Caldera de Taburiente examined in this case study was included in a 1 m resolution DEM covering the whole area where disturbed trees were found and the reaches immediately upstream in the Cantos de Turugumay and Verduras de Alfonso ravines; and in a 5 m resolution DEM for the rest of the case study area (referred to below as MDE1 and MDE5). The altimetric accuracy of the data in these DEM is shown by the RMSE data for each, with an estimated value of z≤ 0.5 m for both models.

To complement this topography, supposedly representative of the basin geometry at the time of the LIDAR survey (2009), a theoretical second representation of the relief was generated, in an attempt to limit possible differences between the date of the LIDAR topography data collection and the date of the event analysed (1997) considering that in an area where there is evidence of bed mobility the twelve year gap between the two dates may have led to variations in the bed topography linked to incision and river sediment processes.

This second topography (referred to below as v1997), compared with the original (v2009), presents surface modifications of the riverbed within the distribution area of disturbed trees. The geomorphological criterion used to include these modifications was as follows: during the initial growth period the trees must have been on ground unaffected by the ravine dynamics and therefore the topographic elevation of the tree base can be used as the maximum elevation limit of the riverbed at that point. To apply this criterion, the v2009 topography was intersected with an inclined surface resulting from the interpolation of the points representing the position of the disturbed trees. This v1997 topography therefore represents the maximum topographic elevation the Taburiente riverbed may have presented at that date.

These topographic models were used to generate two three-dimensional meshes for the area, formed by 1, 3 and 5 m non-structured elements, depending on their location (Figure 5). The smallest elements (1 m) were generated in the area of most



interest, where the disturbed trees are located; the intermediate elements (3 m) were linked to the channel areas upstream and downstream of the area with the disturbed trees; and the largest elements (5 m) on the slopes.

To set the boundary conditions, two independent inflow points were identified, corresponding to the Cantos de Turugumay and Verduras de Alfonso ravines; and a single outflow point was identified in the channel of the Taburiente, approx. 500 m from the area of the disturbed trees.

A dry scenario was also considered as an initial condition for the model since the function of the river is ephemeral, with an estimated basin concentration time of 4.5 hours to its outlet into the sea.

The final feature of the boundary conditions of the model to note is the roughness of the terrain, expressed by the Manning n coefficient. This variable was derived from the photointerpretation of the satellite orthoimages with 0.5 m spatial resolution, and field work. These delimited a series of zones with homogeneous relief roughness, grain size and appearance, and vegetation density, which were then classified and the corresponding Manning's n value (Figure 5) was assigned to them following the relevant tables and graphic guides (Arcement and Schneider, 1989; Brunner, 1995).

Once the hydraulic model was developed, the most important results obtained from model simulation include data related to depth, specific flow, absolute elevation level and velocity field variables. But, just as with the results of the rainfall-runoff model, the absence of flow gauge stations in the basin did not allow calibration of the model and, therefore, the validation of results.

## 4 Results and Discussion

### 4.1 1997 Flood reconstruction

One of the most interesting aspects of the results obtained from the 2D hydraulic modelling carried out in the Caldera de Taburiente National Park is the limited flooded area (very narrow) linked to the modelling of the liquid flows which the rainfall-runoff model obtains for the 1997 analysed event (Figure 6). This statement is based on both a visual analysis of the flash-flood evolution and on the variable 'depth' results obtained at locations assigned to disturbed trees for this event. Thus, for the v2009 topographical model only 6 of the 11 trees give a depth value, and this is a low value, so that the RMSE obtained is around 1 m compared with the height of the identified disturbances. In the 1997 model the number and RMSE are the same as in the previous model, although it is not the same set of trees which provide the depth values.

Based on the above, the peak flows modelled to minimize the RMSE were in all cases greater than those associated with the rainfall which occurred. Thus, as shown in figure 7 (below), compared with scarcely 50 $m^3 \cdot s^{-1}$ obtained as a result of the rainfall-runoff model for the precipitation occurring 11-13 January 1997, the value which minimizes the differences between the flood water depth and the height at which the disturbances were observed is around 1200 $m^3 \cdot s^{-1}$. This important difference noted in the flow values leads to two possible approaches which will discussed below: on the one hand, that the rainfall record for the event analysed is not correct, either because of measurement errors or else because the spatial heterogeneity of the precipitation meant that the amount recorded by the pluviograph was not representative of the rainfall in





the Caldera; and on the other hand, the role of solid (sediment and wood) flow in the flash floods which occurred at the study site.

## 4.2 Statistical approach

### 4.2.1 Maximum precipitation results

The results obtained, from precipitation frequency analysis shown in Figure 8 and Table 1, show how the GEV distribution function systematically underestimates the values of the precipitation quantiles compared with the other two distribution functions, independently of the event duration.

On the other hand, as shown in Table 1, the SQRT-ETMAX function applied to the maximum series yields results higher than those obtained by the GP function applied to the exceedance series, when considering events with duration of fewer

than 24 hours. However, this trend is reversed if events with duration of over 24 hours and medium or high frequencies are analysed.

When high frequency precipitations (2-years return period precipitation, T2) are considered in almost any of these situations, the estimates provided by all the distribution functions may be considered analogous. Based on this convergence of results, there is a strong analytical basis for reporting that the precipitation associated with the 1997 event studied here presents a

low associated return period of around two years. This is interesting as it shows the high recurrence of events which may generate important flash floods, with significant geomorphological and dynamic effects.

### 4.2.2 Flood discharges

Considering the three precipitation distribution scenarios over the three days duration of the event mentioned above allows an examination of the basin behaviour in each case, especially of the aspects related to the ground infiltration capacity in

each scenario and therefore to the surface runoff which produces variations in the flow rates generated. These flow rates (Figure 9 and Table 2) show how the three scenarios considered represent situations which may be more or less favourable to flash flood generation.

A similar pattern can be observed in the generation of peak flows in each scenario for precipitations with duration of under and of over 24 hours, although with some slight differences. Thus, the production of the liquid flow is low when the rainfall

distribution is left-skewed (start of the hyetograph); intermediate when the hyetograph is symmetrical; and maximum when right-skewed. However it can be seen that first, the variations are smaller between the symmetrical and the right-skewed hyetograph than between the symmetrical and the left-skewed hyetograph; and secondly, that in general these differences are significantly reduced when the duration of the precipitation exceeds 24 hours (Table 2).





### 4.2.3 Flood stages

Due to the low correlation between the flow depths associated with the hydraulic model and the height of the tree scars related to this event, hydraulic modelling was run of maximum peak flows to obtain the value minimizing the RMSE between the variable water depth and scar height.

It soon became clear that the peak flow value minimizing the RMSE exceeded the peak flow range modelled in HEC-HMS. To determine this value, a simulation was then run using the 500-year return period value as the second peak flow value. The results of the modelling continued to show a high RMSE and the variable flow depth obtained was always below the variable scar height.

In view of these results, simulation using different peak flow values was continued until the value minimizing the RMSE
was obtained. This value was established (Figure10) as a peak flow of 1235 $m^3$ $s^{-1}$.

### 4.3 Comparing results

The first and most important point to highlight when comparing the results obtained using the hydrological-hydraulic models is the significant difference between the flows obtained from the rainfall-runoff transformation process of the statistical approximation (~50 $m^3$ $s^{-1}$), and those obtained taking the dendrogeomorphological data into account(~1200 $m^3$ $s^{-1}$). This
latter flow value clearly exceeds the return periods considered (maximum 500 years, Table 2) in the rainfall-runoff hydrological model developed for the study area.

In addition, these flow differences mean a distinct behavioural change in this gorge. As shown in Figure 8, it functions either as a braided channel with emerged bars (~50 $m^3$ $s^{-1}$), or as a single channel occupying the whole river bed (~1200 $m^3$ $s^{-1}$).

One of possible reasons for the general non-coincidence between the event dates documented or in the systematic record of
intense precipitations, and those deduced from the dendrochronological analysis of samples, is the non-linear nature of the rainfall-runoff process and its effect on the riparian vegetation.

Thus, the most intense precipitation events in the meteorological station do not necessarily produce the highest flood flow rates. The spatial-temporal variability between precipitation and flow rates is influenced by the local or general character of the maximum precipitation intensities, so that the hydrological response may or may not be limited to one sector of the
basin; and by the ground moisture content previous to the most intense precipitation event, conditioned by the preceding precipitation (previous three days, month or even hydrological year) which strictly controls the runoff threshold and therefore the runoff coefficient.

The source of error and uncertainty deriving from the movable alluvial river bed is linked to solid load transport during flash floods, another feature which complicates or actually prevents flow estimates from palaeostage indicators. In fact, as can be
seen from field-based evidence (crevasse splay deposits, erosive capacity, extremely high palaeostage indicators…), in the torrential phenomena which occur in the Caldera de Taburiente ravines the solid load plays a crucial role, in both the debris bedload (sand, gravel, pebbles, boulders and large boulders), and in the floating load (logs, branches, root systems, etc.).



This transported solid load extensively modifies the flow hydrodynamics, raising the height of the water sheet, incrementing the density and fluid viscosity and therefore its transport capacity with feedback effect; forming pools, narrowing the effective section, concentrating the flow and initiating sudden release of pooled water. Thus, the first level of uncertainty in the event analysis is determining the type of phenomenon, which may range from a rock avalanche to a clear water flood,

including debris flow, hyperconcentrated flow or debris flood; and for the same event, may evolve in space and time through various of these phenomena, which makes interpreting earlier events from the available evidence extremely complicated. The role of the floating woody load is similar to the detritic bedload, leading to important hydrodynamic and turbulent changes and to effects derived from the generation and position of the dendrogeomorphological evidence itself (Ruiz-Villanueva et al., 2013; Ruiz-Villanueva et al., 2014).

**4.4 Flooded areas: past flood event maps**

Combining the use of dendrogeomorphological evidence with detailed hydrological and hydraulic models enabled us to describe the 1997 flash flood event analysed here.

Thus, the peak flow (liquid plus solids) associated with this event was determined. The water sheet area associated with this peak flow was also observed (Figure 8) and was shown to be much more extensive than the area obtained from using only

the meteorological information available in the study area.

This last point is of vital importance in flood risk assessment and for generating the relevant mapping to determine the zones affected or unaffected by flash floods (Figure 10). Thus, the spatial probability of the areas affected when flash floods occur in the zone can be established, taking into account all the uncertainties inherent in the limited data available to compare and contrast the results.

**4.5 Limitations of the data sources used**

Hydrological models such as HEC-HMS are inherently uncertain, due both to the random nature of floods and to the uncertainty component of epistemic origin (due to limited data and knowledge; Ballesteros-Canovas et al., 2013). In this context, model calibration and validation enables an understanding and minimization of the model uncertainty, by the iterative variation of parameter values within a predefined range. However, this proved impossible in this case due to the

lack of observed flow data. As a result, the values assigned to each of the model parameters should be considered as a first approximation, generating some doubt as to whether they adequately reflect the hydrological response of the basin. However, the above limitation is partially mitigated by an in-depth knowledge of the basin physiography and hydrological response. The selected parameter values are therefore within the rainfall - runoff transformation process simulation range at the study site, so that the values assigned are considered consistent.

A second cause of uncertainty is that in all hydraulic models the topography is one of the main and most essential variables in how representative the results really are (Hardy et al., 1999; Omer et al., 2003; Casas et al., 2006). Without topography to represent the particularities of the terrain, the reliability of the results obtained is questionable (Horritt and Bates, 2001a).



Our model is adapted to the conclusions drawn by Casas et al. (2006), in the use of LIDAR data for the topographic representation of the terrain. In addition, the criterion followed for assigning sizes to our 3D mesh optimizes the number of elements in the mesh and is also based on the second conclusion of these authors. Thus, the spatial resolution of the information is another source of uncertainty in the results obtained and is also a factor influencing the different variables used in hydrological and hydraulic modelling (Horritt, 2000; Horritt and Bates, 2001b). Although it has been impossible to eliminate the uncertainty related to this phenomenon, attempts have been made to limit it as far as possible by using the topographic information with the highest resolution available for the area, enabling work in the area of greatest interest with a spatial resolution of 1 m, in both the hydraulic analyses and later in the topographic analysis.

The existing movable bed may imply some uncertainty in the assignment of the roughness coefficient although spatially and temporally most of the channel is occupied by boulder bars. Orthophotos taken at earlier dates, particularly in 2002, show small variations in the extent of the willow forests near the National Park services centre. Analysis of these orthophotos yielded a new map of roughness coefficients. The two-dimensional hydraulic model obtained using this new map does not show significant variations in the depth of the water surface (Figure 7B), so the uncertainty from changes in the values of roughness is negligible.

Another source of uncertainty which cannot be ignored refers to the geographical location of the disturbed trees, as these are an essential feature of the analysis. The trees were located initially through a topographic survey of their positions. However, uncertainties in the measurement and precision of the topographic equipment (Total Station and differential GPS) may have resulted in minimal displacements in the real location of the trees and with the abrupt relief of the area in question this may have led to topographical contour variations which may be important in later studies. To limit this uncertainty when analysing the match between the height of the disturbances and of the water sheet, a 3x3 m buffer was taken into account around the position of the tree (Figure 11). This attempted to visualize the result variability from the effect of this phenomenon, and also to establish the upper and lower limits for the surface adjustments carried out. The results obtained lead to the following conclusion: although the associated RMSE values (Figure 7) vary according to the topographical values used inside the buffer (minimum, maximum and tree location values), this does not modify the return period value associated with the flows which minimize the RMSE, both for the total dendrogeomorphological dataset and for the subsets associated with each of the gorges which drain to the Playa de Taburiente.

The spatial variability of the precipitation should be added to the above as another well-known factor influenced by the terrain orography. Efforts have been underway since the mid-20th century (Spreen, 1947) to understand and quantify the effect of relief on precipitation variability. This problem is common to any drainage basin, and generates uncertainties in rainfall-runoff models when no especially representative record of the basin pluviometry is available. But its effect is magnified in basins with elevated relief and reduced area, where a skewed spatial-altitudinal record may give rise to important under- or over- estimates of the total precipitation in the basin.

A similar approach to the above can be used to analyse the effect of slope orientation on precipitation variability. Ruiz-Villanueva et al. (2013) present the differences in the precipitation curve recorded in a small mountain basin from the effect





of altitude and slope orientation, concluding that these factors play an essential role in the basin precipitation distribution, and that the curves generated (positive with altitude) have a direct influence which explains total precipitation values and gauged flow rates in the basin.

Finally, the uncertainty in the analysis deriving from the scar height and dating data must also be considered. In the Caldera
de Taburiente 63 different wounds were dated corresponding to 8 years but the dendrochronological methods may possibly not have recorded all the flash floods that occurred (Génova et al., 2015). There is no guarantee that all flood events had been dendrogeomorphologically dated (Zielonka et al., 2008; Ruiz-Villanueva et al., 2010) because, e.g. if the wound was old or not extensive, the new tree tissues formed after flash floods may have masked the injuries (Stoffel and Bollschweiler, 2008); or the damaged trees may have disappeared, swept away by flash floods after uprooting (Génova et al., 2015). Two
other sources of uncertainty affect the accuracy of the study: first (and already mentioned above in relation to tree physiology), the measured wound heights in the Caldera de Taburiente correspond to wounds already in process of healing and therefore currently probably lower than when the damage occurred. Secondly, the height was measured from the base of the tree during the fieldwork in 2011-13, 15 years after the 1997 event (Génova et al., 2015). Given the intense flash flood activity on this site, considerable variations in the flow height and direction have occurred in recent years and now the
distance from the base of the tree to the river in 1997 can only be roughly estimated (Génova et al., 2015).

### 4.6 Limitations of the methods used

Flash floods in mountain basins may trigger the mobilization of varying amounts of solid load (i.e. sediment and woody debris), modifying the channel dynamics and morphology (Ruiz-Villanueva et al., 2013). In this respect, the HEC-HMS version here obtains hydrographs that only represent clear water flow as output data. This means that discharges calculated
in this way may be underestimated (Bodoque et al., 2011) and therefore cannot be used to assess risks, since estimated hazard parameters (i.e., depth and velocity) define values lower than they should be. This assertion is borne out by the fact that in physiographic contexts similar to those presented in this paper, sediment load and woody debris may be mobilized together by the water flow. Therefore, flow estimates considering only water flow do not adequately reflect the characteristics of flash floods with low annual exceedance probability.

Another limitation appears with the peak values associated with the synthetic hyetographs generated depend on the torrentiality of the precipitation, for which a regional hourly- daily intensity ratio is used. This regional value may not be appropriate to represent the local torrential fluctuations in the Canary Islands. The effect of this phenomenon would be an underestimation of maximum precipitation values and therefore of peak flows.

On the other hand, applying frequency distribution functions to extrapolate the maximum precipitation values associated
with extreme quantiles (with very low occurrence probability) always generates a degree of uncertainty in the results obtained. This uncertainty (epistemic uncertainty; Merz and Thieken, 2005) is the result of the insufficient length of the data series used as the basis of these predictions; so that the insufficient representativeness of the data series, the loss of high magnitude records, measurement errors, undetected anomalous values, etc., may lead to significant variations in the values





associated with the lowest frequency quantiles. Using threshold exceedance series, thus incrementing the length of the data series and enabling better fit (Cunnane, 1973; Madsen et al., 1997) of the high range of the frequency function, was the method chosen in an attempt to limit the uncertainties linked to these statistical analyses.

With reference to the rainfall-runoff models, the differences in precipitation distribution and peak flow generation in the different models are from the terrain infiltration capacity, maximum when dry. Thus, both when the hyetograph is symmetrical or is skewed at the end of the event, the terrain gradually becomes saturated with low intensity precipitation; and so when the precipitation is most intense, the percentage of precipitation retained by the ground and the vegetation (initial abstractions) has already been reached. This means that the most intense precipitation, the most important in terms of volume, is transformed almost completely into surface runoff and becomes part of La Caldera drainage network flow.

On the other hand, channel erosion-sedimentation processes may lead to topographical variations which may be significant in the results obtained from the hydraulic modelling (Neuhold et al., 2009). As mentioned above, this channel was found to have high sediment mobility and therefore these processes, linked to a single event, may be significant. To evaluate the influence of these processes on the results obtained, the theoretical v1997 topographic surface was defined; only slight variations were observed in the depth values associated with the different flows modelled. It was concluded that for the liquid and solid flow volumes required to cause the tree disturbances observed, the possible changes in the river bed topography associated with incision or deposition processes do not affect the results obtained.

Finally, in relation to the hydraulic modelling, it was decided not to use the turbulence and sediment transport modules as the information required for these was unavailable, including the coefficients defining the turbulent flow behaviour and sedimentograms for sediment incorporation and transport upstream of the study site. Using these modules would therefore not reduce the model uncertainty, and would mean an important increase in computing time.

## 4.7 Validity and extrapolation of the results

We must to keep in mind that not is there necessarily any direct general correlation between the event dates with highest flow rates and the quantity and intensity of dendrogeomorphological evidence. The hydrological flood events best recorded in dendrochronological evidence will, a priori, be precisely of intermediate magnitude (extraordinary), since those of greater magnitude (catastrophic) uproot and destroy all the trees on the valley floor, leaving hardly any evidence; while those of lesser magnitude (ordinary) leave little evidence in number and replication to facilitate detection (Ruiz-Villanueva et al., 2010). Added to this is the temporal proximity of successive events which may mask or screen an earlier or later event of different magnitude, so that e.g. a later event of greater magnitude may destroy all the dendrogeomorphological evidence of another earlier but less intense event, thus eliminating events from the record (Ballesteros et al., 2012); or the later event may be of a higher magnitude meaning neither event is recorded dendrochronologically. This supposes skew requiring interpretation of the different data records event by event, taking into account the time elapsed between them.

However, there is no doubt that the main source of uncertainty in the results is the existence of an alluvial bed in a large sector of the valley bottom in the case study area. This may be considered hydraulically mobile, at least for moderate or high



magnitude events and the finest fractions (gravels and pebbles), and so the riverbed configuration may change significantly from one event to the next, as confirmed from comparing historical series of aerial photos or from multi-temporal field photos. The following are frequently encountered: avulsion phenomena in the main channel, positional changes in flow channels and threads, vertical bed incision, river bank scarp sapping and retrocession from lateral migration of meanders and

braids, bar and island movements, deposits of levees and crevasse splay, etc. In fact, at each point in the history of a flash flood, the effect on the vegetation on the valley bottom may be different depending on the position of the trees, but above all depending on the torrential system dynamics. This, combined with skew from the presence of intensity-based evidence mentioned above, means that the interpretation of flow rate orders of magnitude, levels reached or areas flooded by past events, is still simply a speculative exercise, above all when there is not a significant number of tree replications with

evidence of the same event, homogeneously distributed along the channel.

However, a geomorphological phenomenon has been observed which counteracts this uncertainty related to the movable river bed. There appears to be a compensatory phenomenon of the position of the erosion and deposit zones within the same cross section of the channel; it is as if the material generation and transport in this reach is to some extent in dynamic equilibrium so that the wet section area remains constant over time and what occurs is only that the position of the erosive

and sedimentary forms changes in the cross section. This same effect in a cross section can also be observed to some extent along the Playa de Taburiente reach which acts as a sedimentary basin in equilibrium. This means that the dendrogeomorphological evidence which can be used as palaeostage indicators, such as scars, although they cannot be used in this case study for a precise flow estimate; give an idea of the order of magnitude of the flow volumes (water plus sediment) over time for a given section.

Finally, even if all the uncertainties and error sources mentioned above are resolved, and each past event is successfully linked with a date of occurrence and magnitude (normally flow rate), including it as non-systematic data in the statistical flow rate analysis is complicated. As well as the usual difficulties deriving from the inclusion of historical data in statistical analyses (Francés, 2004), this case study adds the non-availability of systematic flow data, since the only possible gauge on the main river (Azud de derivación del tomadero de Dos Aguas), simply records ordinary flow rates or low magnitude

floods; for other events this diversion dam leaves the flow ungauged to the river mouth.

All these problems and uncertainties, which condition the validity of the results obtained and their application, are not exclusive to the case study of this reach in the Caldera de Taburiente National Park, but on the contrary are common to all gorges and ravines in the Canary Islands (with limited or no gauging), to most of the Macaronesian archipelagos and to many mountain basins in the Iberian Peninsula, especially ungauged reaches. For this reason, confronted with this lack of

gauge data or the possibility of obtaining quantiles from calibrated rainfall-runoff models, the information on flood flow frequency derived from dendrogeomorphological evidence is shown as an approximation, and even though with limited statistical reliability is justified as the only evidence objectively available and therefore useful for flood risk and hazard assessment.



Thus it is obvious that to be able to reproduce the analysis performed here is a priority aim, so that this can be used as an intrinsic control factor for the proposed methodology. This can be evaluated from two different points of view: from a spatial viewpoint the methodological approach in this study can be easily reproduced in other regions with similar conditions (small ungauged mountain basins), taking into account that even with the limitations of this approach, this methodology gives more

reliable results than non-calibrated and non-validated hydraulic analysis. In this case, the results obtained from the dendrogeomorphological analyses provide a model calibration source.

On the other hand, from a time-based viewpoint, the methodology applied here can be reproduced for past situations whenever terrain-specific information is available at the relevant point in time, although here problems may occur to limit its application to past events. Two variables affect the applicability of the proposed method: the level of vegetation cover and

the possible species variations, spatial distribution or growth level, factors which control the terrain roughness and therefore the result of the hydraulic models (Casas et al., 2010); and secondly, variations in basin morphology over time, and therefore the influence of incision and deposit processes on the geometrical configuration of the channel. While the first of these problematic variables has been defined to a certain degree since the mid-20th century (using available information from the earliest aerial photograms of the area), the second variable cannot be solved with the spatial resolution or scale of historical

topographic maps, incompatible with the generation of detailed hydraulic models.

### 4.8 Future prospects

Reconstructing flow rates of past events in an alluvial bed channel with such important solid load transport can only be adequately simulated using: a two-dimensional movable bed hydraulic model with bedload transport and turbulence, such as the Iber software with all modules activated (Bladé et al., 2014); high resolution spatial topography for the channel geometry

such as that derived from periodical LIDAR surveys; tachymetric surveys in particular sections and slope breaks; input diameter values derived from three-dimensional laser scanner analysis complemented with field calibrations to estimate the bed granulometry (Vericat et al., 2006; Brasington et al., 2012).

Another interesting future possibility would be to implement modules to consider the role of the floating solid load transported (woody material) such as the Woody Iber model software (Ruiz-Villanueva et al., 2014). For this, a quantitative

estimate is required of the incorporation of this woody load into the channel using different simulation scenarios (Ruiz-Villanueva et al., 2013).

Finally, other types of dendrogeomorphological evidence can be used in case studies, including tilted trees which are now being used to estimate past event magnitudes (Ballesteros-Canovas et al., 2014).



## 5 Conclusions

The combined and successive use of systematic meteorological and dendrogeomorphological data sources with hydrological analysis, rainfall-runoff models and two dimensional hydraulic models have enabled the detection, dating and assigning of magnitudes and frequencies to past flood events in the Playa de Taburiente.

It is only with the results of this palaeo-hydrologíguc-hydraulic research, combined with meteorological analysis of the systematic record, that some basic recommendations can be made on the hazard and risk of torrential flash floods and their associated flooding which along with exposure and vulnerability analysis will help managers to manage and minimize the risk of torrential flash floods in the National Park.

**Acknowledgements**

This study was funded by the MARCONI research project (CGL2013-42728-R) and MAS Dendro-Avenidas research project (CGL2010-19274; www.dendro-avenidas.es), Spanish Ministry of Economy and Competitiveness; and by the IDEA-GesPPNN project (OAPN 163/2010; www.idea-gesppnn.es) funded by the Spanish Ministry of Agriculture, Food and Environment (National Parks Research Program. The authors wish to acknowledge the collaboration of other members of the research teams of three projects; the Director (Angel Palomares), technicians and rangers of the Caldera de Taburiente

National Park; and the Geographic Institute of Spain (IGN and CNIG; Juan Carlos Ojeda and Juan Sagües) for the transfer of information and LIDAR images.

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





| < 24 H. RAINFALL EVENTS | | | | | |
|---|---|---|---|---|---|
| T (yr) | p(T) | SQRT-ML [1] | GEV-LMOM [1] | GEV-PWM [1] | GEV-MOM [1] | GP-POT [2] |
| 2 | 0.5 | 145.0 | 147.5 | 147.0 | 151.0 | 156.2 |
| 5 | 0.8 | 234.0 | 223.7 | 223.0 | 226.0 | 224.4 |
| 10 | 0.9 | 304.0 | 276.5 | 277.0 | 275.0 | 275.3 |
| 25 | 0.96 | 402.0 | 346.1 | 347.0 | 335.0 | 349.0 |
| 50 | 0.98 | 482.0 | 399.8 | 403.0 | 379.0 | 410.7 |
| 100 | 0.99 | 569.0 | 454.9 | 460.0 | 422.0 | 480.0 |
| 200 | 0.995 | 662.0 | 511.7 | 519.0 | 465.0 | 551.1 |
| 500 | 0.998 | 793.0 | 589.7 | 600.0 | 520.0 | 660.3 |

| > 24 H. RAINFALL EVENTS | | | | | |
|---|---|---|---|---|---|
| T (yr) | p(T) | SQRT-ML [1] | GEV-LMOM [1] | GEV-PWM [1] | GEV-MOM [1] | GP-POT [2] |
| 2 | 0.5 | 367.0 | 293.6 | 297.0 | 300.0 | 278.1 |
| 5 | 0.8 | 472.0 | 441.4 | 445.0 | 442.0 | 482.3 |
| 10 | 0.9 | 637.0 | 523.1 | 528.0 | 519.0 | 658.6 |
| 25 | 0.96 | 872.0 | 610.8 | 618.0 | 600.0 | 942.6 |
| 50 | 0.98 | 1069.0 | 666.4 | 676.0 | 650.0 | 1208.7 |
| 100 | 0.99 | 1274.0 | 714.6 | 726.0 | 693.0 | 1531.2 |
| 200 | 0.995 | 1509.0 | 756.7 | 770.0 | 730.0 | 1922.9 |
| 500 | 0.998 | 1831.0 | 804.3 | 821.0 | 771.0 | 2573.9 |

**Table 1. Rainfall quantiles obtained by fitting different frequency distribution functions to both, (1) annual maximum series (SQRT-ETmax; GEV), and (2) exceedances over the threshold series (GP).**



| PEAK FLOOD (m³/s) | | | | | | | | |
|---|---|---|---|---|---|---|---|---|
| Return Period T (yr) | | POINT 1 | POINT 2 | POINT 3 | POINT 4 | POINT 5 | POINT 6 | POINT 7 |
| 1997 RAINFALL EVENT | LS | 22.1 | 16.4 | 43.7 | 41.8 | 61.8 | 129.6 | 143.9 |
| | S | 24.1 | 18.1 | 47.9 | 49.4 | 69.7 | 150.3 | 167.5 |
| | RS | 25.1 | 19.0 | 50.5 | 51.5 | 73.7 | 158.3 | 176.4 |
| T2 | LS | 12.9 | 9.2 | 24.5 | 23.2 | 35.0 | 75.6 | 83.4 |
| | S | 37.3 | 26.9 | 71.9 | 60.2 | 99.0 | 199.6 | 219.3 |
| | RS | 47.2 | 35.3 | 94.9 | 85.3 | 133.8 | 275.6 | 304.6 |
| T5 | LS | 26.8 | 19.0 | 51.4 | 49.0 | 72.4 | 155.8 | 172.9 |
| | S | 63.6 | 47.2 | 126.2 | 116.6 | 176.4 | 367.5 | 407.8 |
| | RS | 74.3 | 56.7 | 151.8 | 146.5 | 216.6 | 456.8 | 509.8 |
| T10 | LS | 39.4 | 28.0 | 75.8 | 72.1 | 106.2 | 227.2 | 252.8 |
| | S | 83.6 | 62.8 | 168.4 | 162.1 | 236.8 | 499.3 | 557.0 |
| | RS | 94.4 | 72.6 | 194.2 | 193.1 | 278.0 | 592.7 | 664.0 |
| T25 | LS | 59.3 | 42.8 | 115.4 | 110.1 | 160.7 | 343.4 | 382.8 |
| | S | 112.6 | 85.6 | 229.7 | 229.8 | 325.9 | 692.2 | 775.8 |
| | RS | 123.3 | 95.4 | 255.1 | 260.8 | 367.9 | 789.4 | 886.7 |
| T50 | LS | 76.9 | 56.1 | 150.8 | 144.9 | 210.0 | 448.8 | 501.5 |
| | S | 137.0 | 104.8 | 281.6 | 288.2 | 401.7 | 857.7 | 964.0 |
| | RS | 147.4 | 114.4 | 305.8 | 317.4 | 442.8 | 953.3 | 1072.4 |
| T100 | LS | 98.0 | 71.7 | 193.7 | 186.7 | 267.8 | 573.8 | 642.6 |
| | S | 164.1 | 126.2 | 339.5 | 353.8 | 486.3 | 1043.6 | 1177.1 |
| | RS | 174.2 | 135.6 | 362.3 | 380.6 | 526.6 | 1136.6 | 1279.7 |
| T200 | LS | 120.5 | 88.2 | 239.1 | 231.8 | 330.1 | 706.3 | 792.6 |
| | S | 191.8 | 148.1 | 399.1 | 421.3 | 573.0 | 1235.1 | 1395.8 |
| | RS | 201.6 | 157.3 | 420.4 | 445.2 | 612.5 | 1323.3 | 1491.3 |
| T500 | LS | 155.9 | 115.1 | 310.8 | 304.2 | 428.5 | 919.6 | 1033.2 |
| | S | 234.2 | 181.6 | 489.8 | 525.0 | 705.6 | 1528.4 | 1730.8 |
| | RS | 243.5 | 190.4 | 508.8 | 543.8 | 743.4 | 1607.8 | 1813.6 |

**Table 2. Rainfall-runoff model peak flows in Taburiente fluvial basin location points associated with both, January 11-13, 1997 registered precipitation, and GP (POT) rainfall quantiles. For each of these points of interest, the peak flow obtained for the three analyzed hyetograph models (symmetrical, left skewed and right skewed) is shown. Points 1 and 2 represent the input flow to the "Playa de Taburiente" by "Cantos de Turugumay" and "Verduras de Alfonso" ravines, respectively. Point 3 is located at the outflow point of the "Playa de Taburiente".**



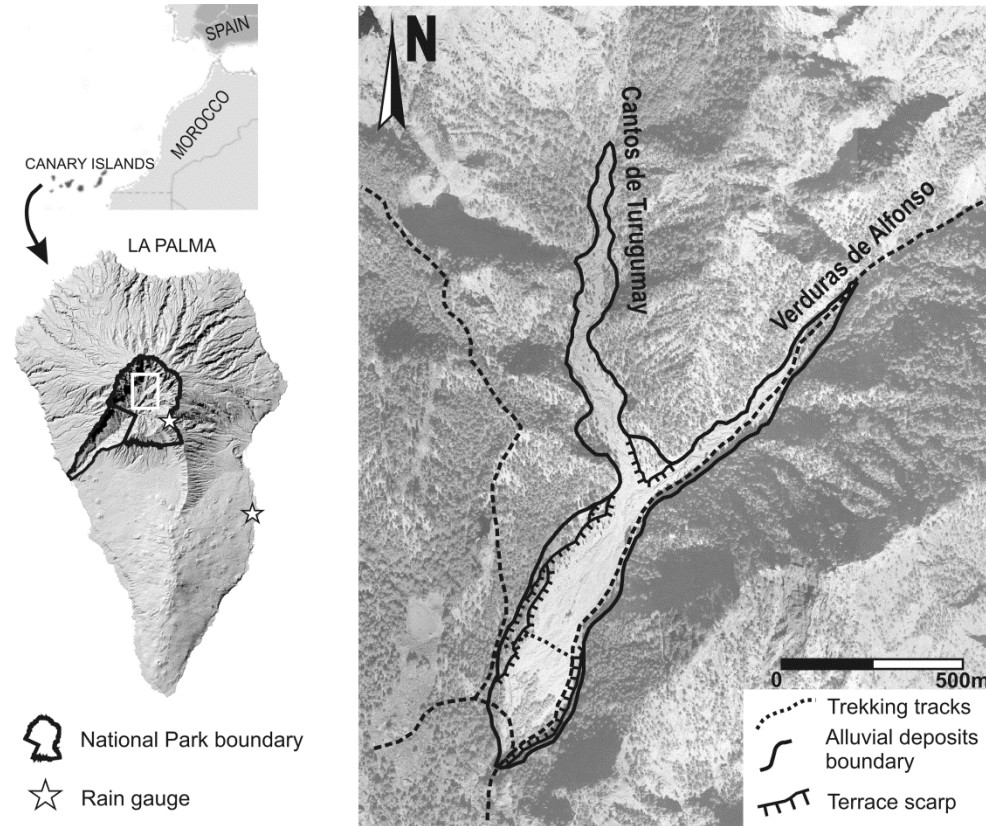

**Figure 1: Location map of the "Playa de Taburiente" area, within the limits of the Caldera de Taburiente National Park; as well as its main elements of interest.**



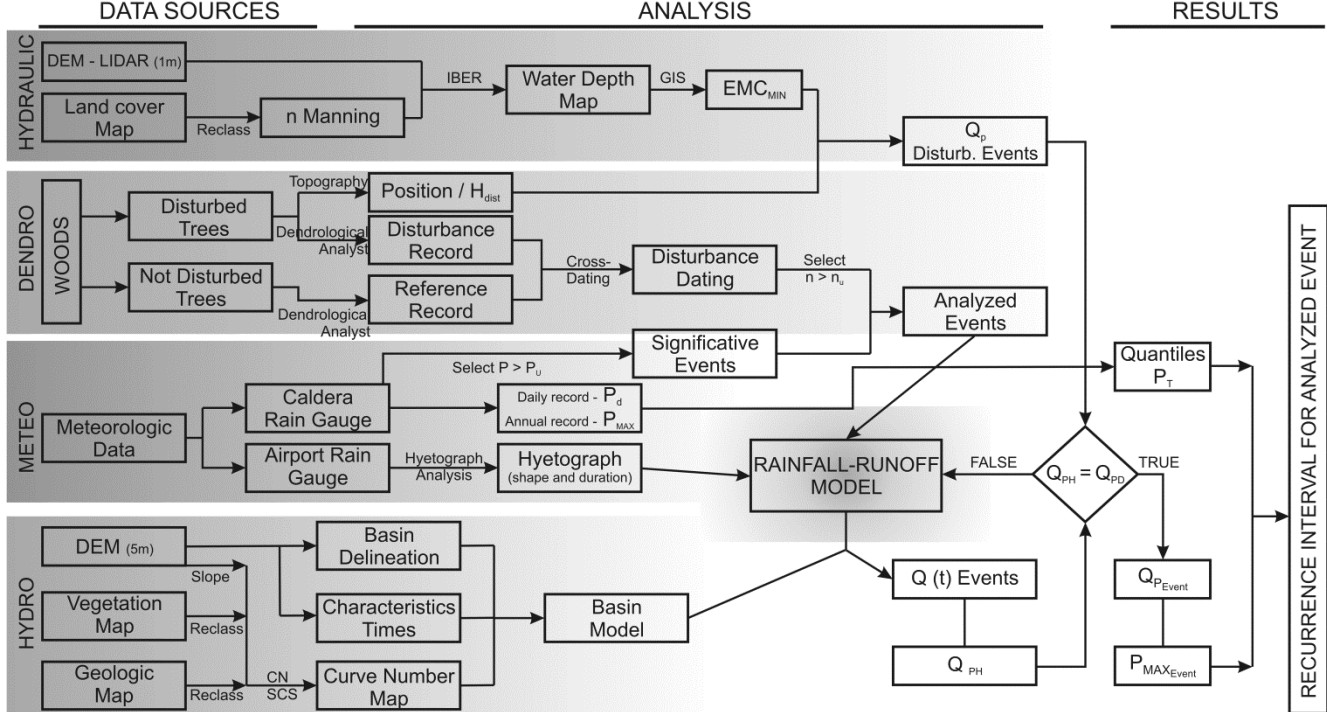

**Figure 2. Representative flow diagram of both, data sources and analysis procedures and established relationships between them, designed to estimate the hydraulic characteristics of palaeoflood events within the Caldera de Taburiente National Park.**



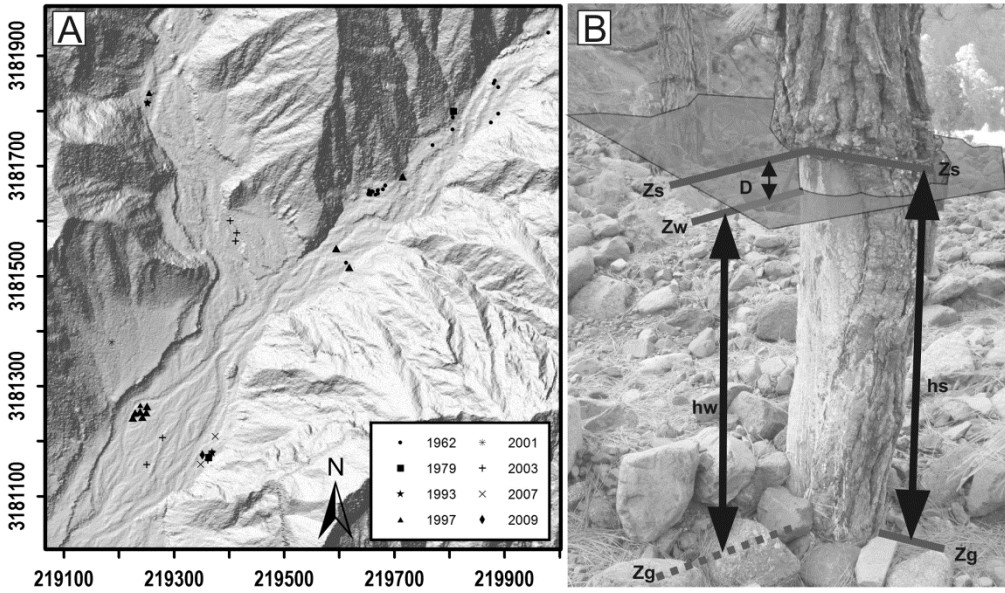

**Figure 3. "Playa de Taburiente" dendrogeomorphological evidence location and distribution map (A). Description of the heights and levels of interest in the study (B): Zs, scar topographic elevation; Zw, water topographic elevation; Zg, ground elevation; hs, scar height; hw, water depth; D, height difference between scar and water surface (deviation).**





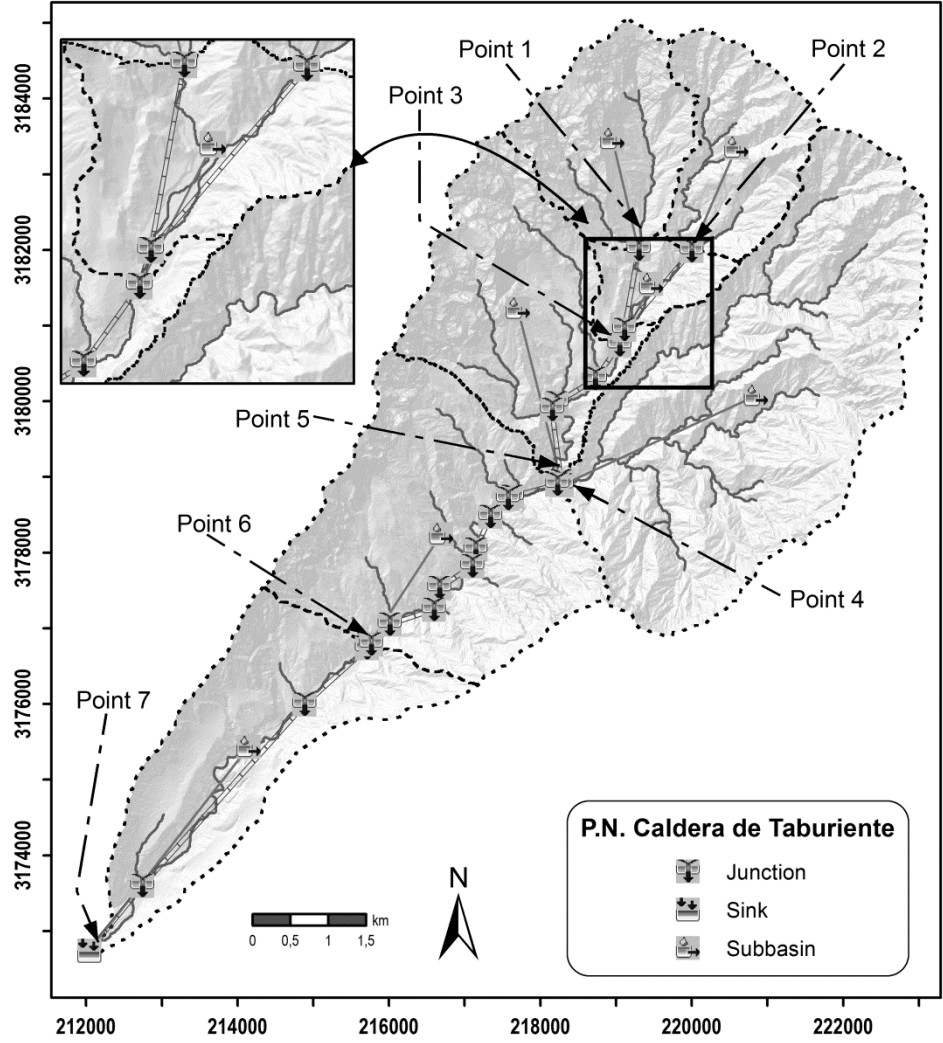

**Figure 4. Developed rainfall-runoff model (HEC-HMS) for Taburiente river basin, with its main features: sub-basins, drainage network, junctions, propagation reach, etc ... (A). B, detailed view of the model in the "Playa de Taburiente" area. Points 1 to 7 show the location of peak flow data of Table 2.**





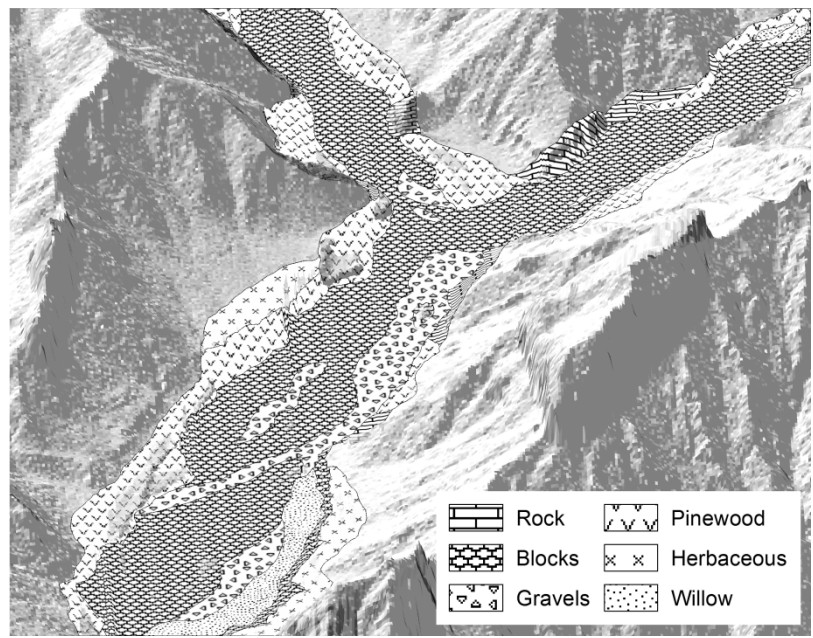

**Figure 5. Generated 3D mesh for hydraulic simulation of floods (Iber software). Terrain roughness (n Manning coefficient) is overlapped to mesh to allow identification of homogeneous areas. Differentiated classes and corresponding roughness coefficients are: bedrock (n=0.4), boulder bar (n=0.055), gravel bar (n=0.035), pine forest on boulders (n=0.05), grassland (n=0.035), willow forest (n=0.07) and disperse vegetation (n=0.06).**




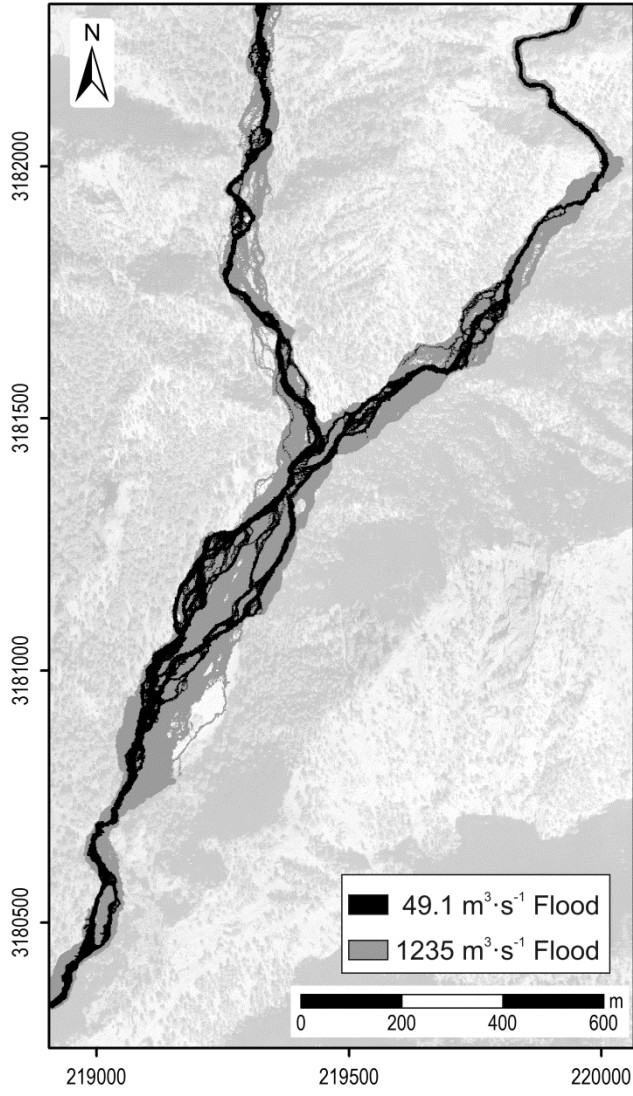

**Figure 6. Hydraulic model outputs. Water surface extension associated both, January 11-13, 1997 precipitation event (49.1 m3·s-1), and liquid flow rate (1235 m3·s-1) that minimizes the RMSE in the comparison between the heights of the water surface and dendro-geomorphological evidence.**

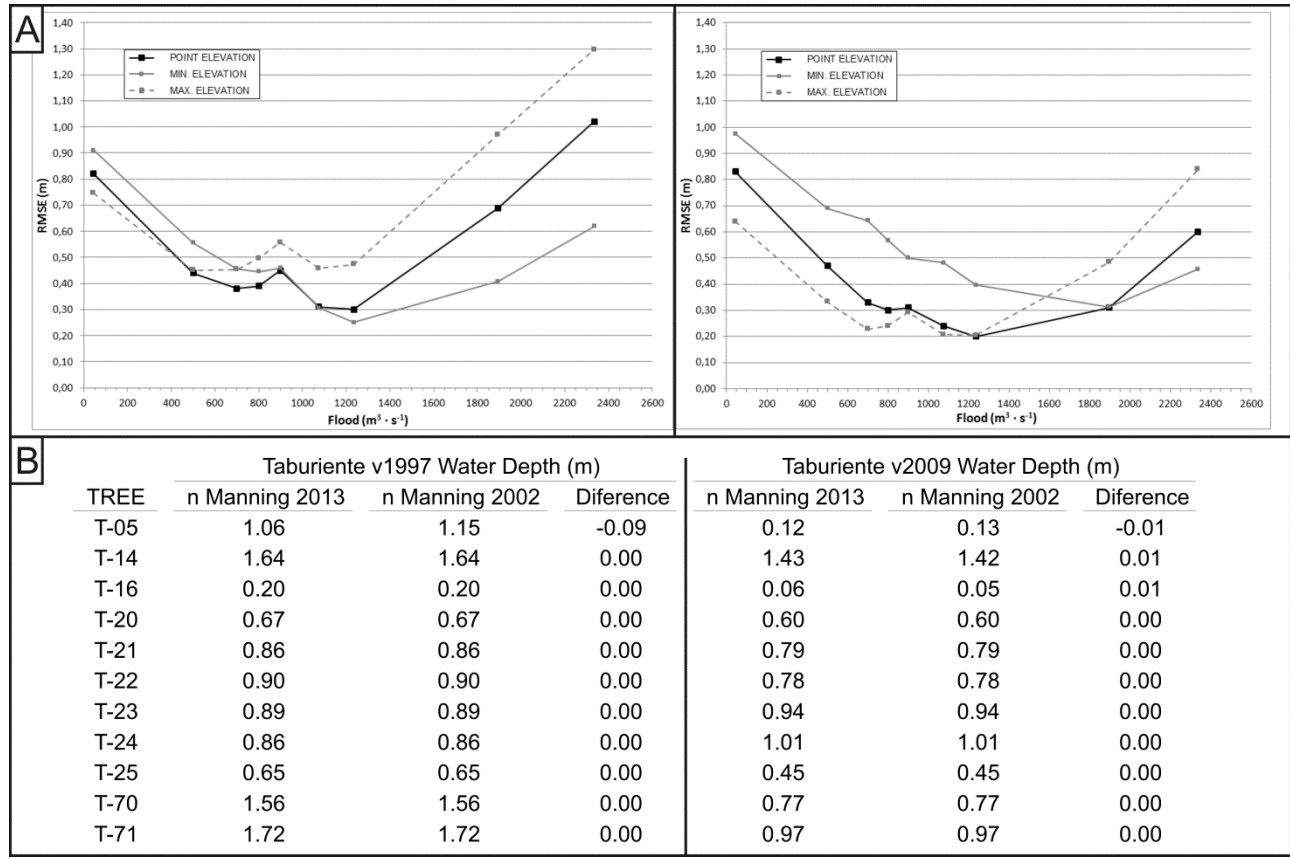

**Figure 7. A,** RMSE values for different modeled flows in comparison with the heights of the dendro-geomorphological evidence related to 1997 flood event: results from actual topography scenario (left), and results from estimated maximum topographic surface at event date (right). **B,** difference in water depth on tree location due to surface roughness change.





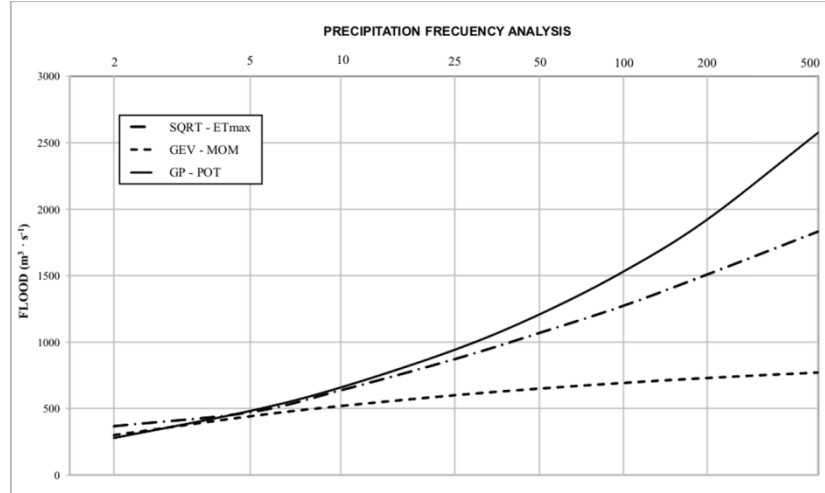

**Figure 8. Results for rainfall frequency analysis in the study area. SQRT-ETmax and GEV functions were applied to annual maximum data series, while the Generated Pareto (GP) function has been applied to exceedances over threshold data series (POT technique).**





**Figure 9. Rainfall – runoff graphic for the three simulated rain scenarios: left skewed (A), symmetric (B) and right skewed (C). The different hydrologic response of the basin in the runoff generation is shown.**





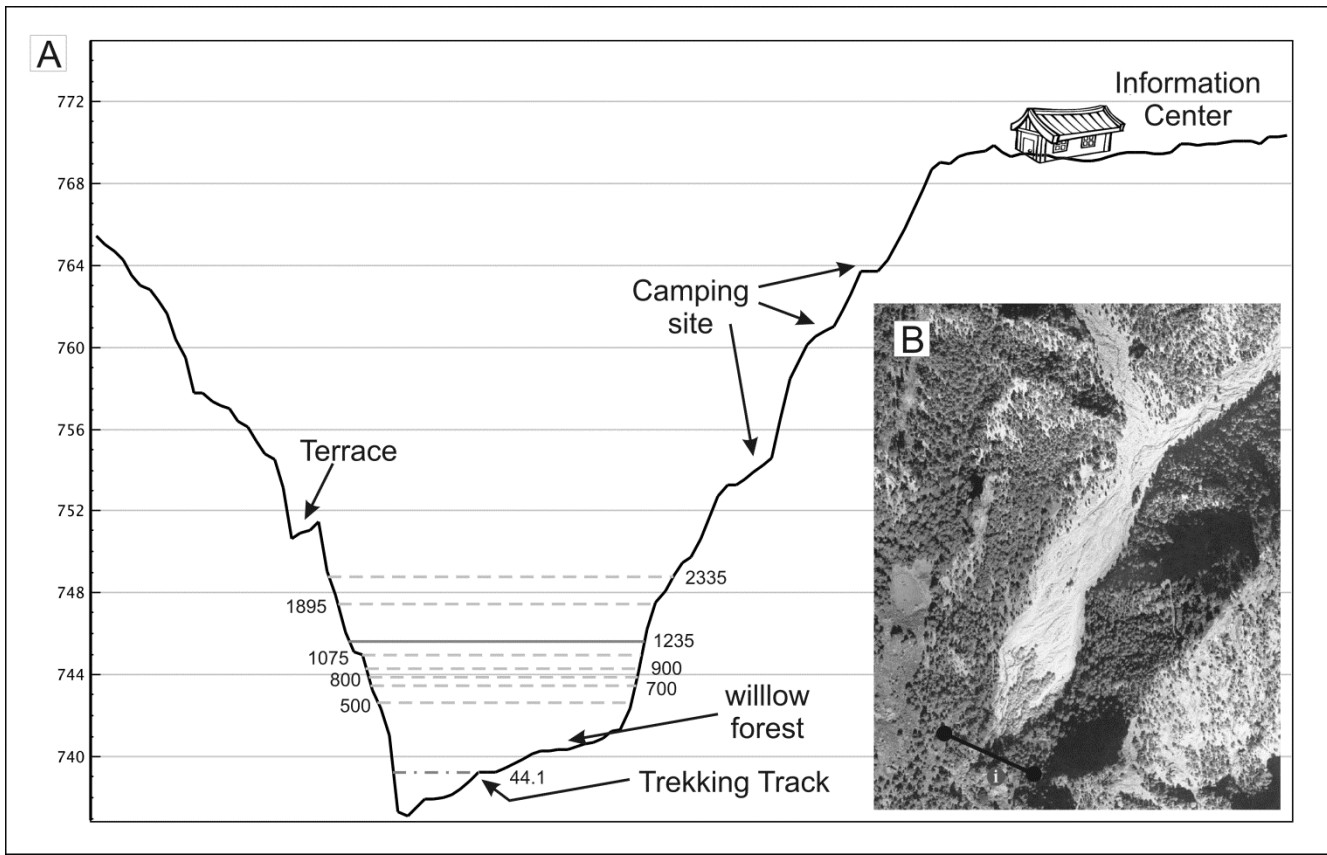

**Figure 10. Flow – flood stage graphic near the Information Centre of the Caldera de Taburiente National Park. Shown (A) flood stage associated recorded rainfall during 1997 event (dashed, dot - line); flood stage for RMSEmin relative to water surface – scar evidence height (solid line); and other modeled flows (dashed line). B, location of the cross section associated with flood stage values, and Caldera de Taburiente National Park Information Center.**



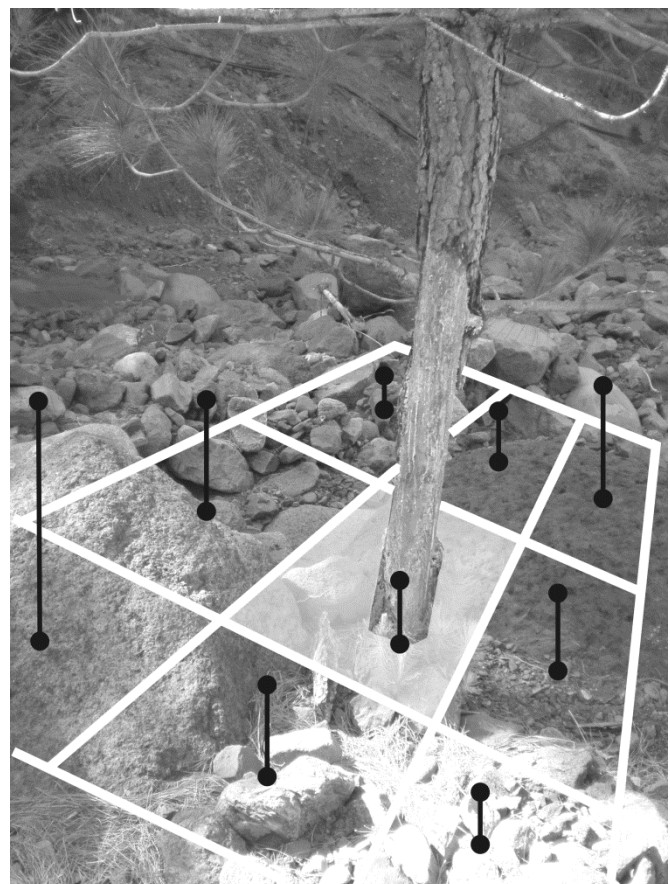

**Figure 11. Sketch for methodological approach followed for the sensitivity analysis of the flow estimation results against possible variations in the location of dendro-geomorphological evidence. Red lines show the difference in ground height inside the buffer area.**