# Peer review of "Nat. Hazards Earth Syst. Sci. Discuss., doi:10.5194/nhess-2016-206, 2016 Manuscript under review for journal Nat. Hazards Earth Syst. Sci. Published: 20 June 2016"

_Natural Hazards and Earth System Sciences, 2016_

## Referee Comment (RC1) · Anonymous Referee #1 · 17 Aug 2016

The paper presents an interesting case study. In the process of reconstructing the characteristics of a flash flood event, rainfall information and proxi-information (dendrogeomorphological) provide highly inconsistent results. The flood peak discharge estimated by rainfall runoff modelling is more than one order of magnitude lower than the one obtained by fitting the water stages to the scars of trees with an hydraulic model. The paper rightly suggests that there is a benefit in performing analyses with different data since the contrasting results warn us on the fact that some more analysis

is necessary to find out what really was the order of magnitude of the flood peak.

Despite the interesting case study, I do not like the paper and recommend rejection for the following reasons:

- I do not think that the statistical analysis of rainfall data is correct. I am surprised that the GEV and POT-GP method give such different results. In theory they should be analogous (being a Pareto distribution for peak-over-threshold mathematically correspondent to a GEV distribution for annual maxima). Is it because of the method used to estimate the parameters? I strongly recommend to add the plotting position representation of the data in figure 8 (which, by the way, should have rainfall and not flood peak in the y-axis). This would show where the estimation problem has gone wrong.

- I do not agree with the strategy of using over sofisticated methods when order of magnitudes are of interest. For instance, in section 3.2.2 the method for modelling the routing of the flood wave is presented, which require LIDAR data for the channel morphology. This is coupled with a very crude ranoff generation model (SCS-CN method + unit hydrograph with parameters estimated from empirical formulae) forced by highly uncertain rainfall inputs. What is the rationale for doing this? Much better would be to use simple models and account for the uncertainties involved. I strongly disagree with the "Future prospect" section of the paper, which suggests to use more complex models in the future. This will add other uncertainties and will not solve those discussed within this paper.

- The title and abstract state that a frequency analysis of (8?) past flash-flood events is performed. In reality only one event is analysed and the probability of exceeding such an event has not been estimated. The "Conclusion" section does not say what the paper contains.

- The methodologies used have not been presented clearly. For instance, what is the "SQRT-ETMAX method"? What is the "RMSE over disturbed tree sample"? What model has been used to generate synthetic hyetographs? How have the daily precipi-

tation observations been used to constrain the generated hourly timeseries?

---

## Referee Comment (RC2) · Anonymous Referee #2 · 18 Aug 2016

Summary of the paper: In this paper an approach for the assessment of past flash-floods in ungauged basins based on dendrogeomorphological information is presented. The used dendrogeomorphological parameters are wounds in trees (scars) resulting from currents in combination with sediments. The authors set up both a rainfall-runoff-model and a 2D-hydrodynamic numerical model. In a case study for the 1997 flood, they make two simulations: (1) A model run with the rainfall-runoff-model with precipitation data from the 1997 flood event. This results in a peak flow of about

50 m$^3$/s. (2) They used the 2D-model to analyse which peak flow matches best to the found tree wounds dedicated to the 1997 flood. This results in a peak flow of about 1200 m$^3$/s. The authors also make an extensive discussion on uncertainties in their analyses.

General/major comments: Generally, the paper addresses an important field of hydrology, namely the assessment of floods in ungauged basins. The data availability in such regions is poor and additional information help to understand the characteristics of the basin and the river system. An interesting method to gain additional information is using dendrogeomorphological data sources. However, there are some weak points in the paper, which should be improve before publication: (1) Since the main focus of the paper is on the use of dendrogeomorphological data sources, there is a lack of explanations in the methods section, how the dendrogeomorphological data sources are obtained in detail. Later in this paper the uncertainties in the data become very important. Hence, the methods section should include more information on how wounds in trees (scars) are generated (only by currents or in combination with sediments); how the time-dependent development of wounds can be described; how wounds are defined as significant; do the data only result from visual inspections; what is the data quality etc.. (2) A vital result of the paper is the large discrepancy in the model runs resulting in 50 m$^3$/s and 1200 m$^3$/s. There is a very extensive discussion on uncertainties in data and methods, which is honorable. From my point of view the discrepancy of the results is that large so talking about "uncertainties" is critical. Isn't it more a signal, that either data or methods are simply not suitable to answer the research question? Where is the border between uncertainty and infeasibility? Especially since both precipitation and dendrogeomorphological data include major uncertainties the conclusion that the obtained results could improve flood hazard and risk analysis is questionable.

Minor comments: P5, L6: Include more information on dendrogeomorphological data and methods (see above). P6, L10: The POT extreme value statistics is based on data lasting <24 hours and >24 hours. Please explain this more in detail. From my point

of view one has to choose defined duration levels (e.g. 6 hours, 12 hours etc.) and perform the statics for each series individually. P7, L24: Please explain more in detail how the v1997 topography was built. P10, L11: The explanations given in sec. 4.3 can only hardly justify this large discrepancy found in the model results. Figure 8: Please add the original data (plotting positions) in the plot; the differences in the results seems a little bit strange. Please check also the whole calculation.

Final recommendation: Considering the comments above I recommend publication of the manuscript in NHESS after major revision.

---

## Author Comment (AC1) · 29 Sep 2016

First, the authors would like to thank the anonymous reviewer for his/her suggestions and comments, which have helped us improve our work. However, the authors disagree in some cases with the reviewer's comments and below we will attempt to respond to each issue.

1- I do not think that the statistical analysis of rainfall data is correct. I am surprised

[Figure]

that the GEV and POT-GP method give such different results. In theory they should be analogous (being a Pareto distribution for peak-over-threshold mathematically correspondent to a GEV distribution for annual maxima). Is it because of the method used to estimate the parameters? I strongly recommend to add the plotting position representation of the data in figure 8 (which, by the way, should have rainfall and not flood peak in the y-axis). This would show where the estimation problem has gone wrong.

In the case of the statistical analysis of rainfall data presented in the paper, the GEV distribution was fitted using three different methods: moments, probability weighted moments, and maximum likelihood. The SQRT-ET max distribution was fitted using the maximum likelihood method and the GP-POT distribution was fitted using the L-moments method. It is also worth mentioning that SQRT-ET max distribution (Etoh et al., 1986) was proposed by the Spanish Government in the manual "Maximum daily rains in continental Spain, (Ministry of Civil Works, 1999)", and has been used in numerous scientific studies conducted in Spain (for example, Ruiz-Bellet et al. 2015, Bianucci et al. 2013, Bianucci et al. 2015, or Rico et al. 2001). One of the reasons given in this manual for choosing the SQRT-ET max distribution (as opposed to others such as GEV, TCEV, and LP3) was that it provided more conservative results than other distributions. In our case, this claim showed to be true, with both functions providing similar values over a return period of 25 years and, beyond that, the SQRT- ET max distribution actually giving more conservative results. It is however true that this behavior does not occur when comparing the results shown by the GEV and SQRT-ET max distributions. From the above, and as the reviewer proposes, a full review of the statistical analysis conducted will be provided, with special emphasis on the results shown by the GEV distribution.

2- I do not agree with the strategy of using over sofisticated methods when order of magnitudes are of interest. For instance, in section 3.2.2 the method for modelling the routing of the flood wave is presented, which require LIDAR data for the channel morphology. This is coupled with a very crude ranoff generation model (SCS-CN method

[Figure]

+ unit hydrograph with parameters estimated from empirical formulae) forced by highly uncertain rainfall inputs. What is the rationale for doing this? Much better would be to use simple models and account for the uncertainties involved.

The authors also share the view that the estimation of uncertainties is an important aspect. However, we do not believe that the use of simpler models is much better. Therefore, our approach was to try to use the best possible models conditioned by the availability and quality of data. In short, where the input data allow, more sophisticated methods will be used to avoid the accumulation, at least in these processes, of errors and uncertainties. For example in the case of the hydraulic model, we consider the use of a high detailed DEM to represent as closely as possible the actual ground surface morphology, which improves the accuracy of the results. For this reason, and given the availability of data, we chose to use LIDAR topographic data within a 2D hydraulic model. The choice of this type of hydraulic model was based on its greater ability to reproduce the complex movement of water in these kinds of avenues. With regard to the rainfall-runoff model, we chose the simulation methods for the various processes involved based on the information available for the basin. The choice of the SCS-CN loss method, which is well-developed and widely accepted in the United States and abroad (Ponce and Hawkins, 1996), was based, on the one hand, on the availability of high resolution orthophotographs for the area and, on the other, on the possibility of conducting fieldwork to test and adjust the values previously obtained by photo-interpretation. Thus, input data concerning the CN value was considered as representative of the basin characteristics. The use of other methods of estimating initial losses or abstractions (like Green-Ampt) was discarded due to the lack of more detailed soil composition and textural data. Finally, regarding the transformation method (Clark Unit Hydrograph), the absence of rainfall and flow data associated to the same event reduces the availability of transformation methods in the HEC-HMS environment to only two options. With regard to the consideration of the associated uncertainties, although we considered this to be an essential point, the absence of flow data to calibrate and validate the hydrological output model flows makes it impossible to estimate

this in each hydrological calculation and its overall effect as the methodology unfolds.

3- I strongly disagree with the "Future prospect" section of the paper, which suggests to use more complex models in the future. This will add other uncertainties and will not solve those discussed within this paper.

In this case, the authors do not agree with the reviewer's opinion. The authors do not propose a move towards more complex models, but toward models that attempt to reflect the complexity of flash-flood processes. Undoubtedly, a degree of uncertainty exists for the results of any model with regard to the actual behavior of the modeled phenomenon. Efforts to make the starting point of the model as close to reality as possible should be one of the ultimate goals. The use of clear water along with a low-resolution DEM could be useful in the hydraulic modelling of the slow onset floods typically found in main rivers and in non-urban areas. However, for flash floods in both urban areas and small mountain basins, previous studies have shown that more simple hydraulic models do not work correctly. 1D hydraulic models are not able to solve the complexity of this type of floods, and 2D hydraulic models provide better results (for example: Costabile et al. 2015, Ernst et al. 2010, o Mignot et al. 2006). Clearly, further work is required to better understand the behavior of small, mountain basin flash floods in order to gradually reduce the uncertainties associated with each process. However, as shown in this work with the use of dendrogeomorphological evidence, these uncertainties can also be reduced by applying proxy data from floods that have occurred in the past.

4- The title and abstract state that a frequency analysis of (8?) past flash-flood events is performed. In reality only one event is analysed and the probability of exceeding such an event has not been estimated. The "Conclusion" section does not say what the paper contains.

Actually, in the Abstract the authors state that dendrogeomorphological data is available for eight different flood events. But later, in the same abstract, the authors focus

the paper on the 1997 flood event. This discrepancy is due to the availability of sufficiently replicated and representative dendro evidence for the study. Thus, from the eight different flood events with dendrogeomorphological evidence, only the data from the 1997 event has a wide enough spatial distribution to be valid for comparison with the results of hydraulic modeling. In the other flood events, or the amount of dendro data was not enough (few replications) or the spatial distribution was highly skewed and, consequently, it was not considered valid for comparison with the results from the hydraulic models. Be that as it may, the Abstract will be reviewed to try to rectify any confusion in the interpretation of the analysis carried out.

5- The methodologies used have not been presented clearly. For instance, what is the "SQRT-ETMAX method"?

This is an extreme value distribution proposed by Etoh et al. (1986), which has been broadly used in Spain for annual maximum rainfall data statistical analysis. SQRT distribution has been used to calculate the daily maximum rainfall in Spain for different return periods (Spanish Civil Work Ministry, 1999). Several studies regarding SQRT distribution and its application in Spanish rainfall series have been done, for example: Ferrer, 1996; Zorraquino, 2000.

6- What is the "RMSE over disturbed tree sample"?

This value refers to the fit error between the heights of the nibs (dendrogeomorphological evidence, FDES) and the water surface height associated with flood modeling in that location.

7- What model has been used to generate synthetic hyetographs?

The synthetic hyetograph has been built using the alternating block method (Chow et al. 1994).

8- How have the daily precipitation observations been used to constrain the generated hourly timeseries?

[Figure]

Distribution of daily rainfall in the form of hourly time series has been done using the IDF curves currently used in Spain for the study area (Salas and Fernandez, 2007). The procedure for obtaining these IDF curves is described in works like Salas and Fernandez-Yuste (2006a and 2006b).
* * *

---

## Author Comment (AC2) · 29 Sep 2016

First, the authors would like to thank the anonymous reviewer for the suggestions and comments provided, which will help improve our work. Below we respond to his/her comments.

1- Since the main focus of the paper is on the use of dendrogeomorphological data sources, there is a lack of explanations in the methods section, how the dendrogeomorphological data sources are obtained in detail. Later in this paper the uncertainties in the data become very important. Hence, the methods section should include more information on how wounds in trees (scars) are generated (only by currents or in combination with sediments); how the time-dependent development of wounds can be described; how wounds are defined as significant; do the data only result from visual inspections; what is the data quality etc..

We agree with the reviewer regarding the importance of using dendrogeomorphic data in our work. In this regard, the authors did not consider necessary the inclusion of a more detailed description of the characteristics of this data, since this information can be found in Genova et al. (2015), as we explicitly mentioned in the text. However, if the reviewer considers it necessary, the description of the data acquisition method, its characteristics, and the use of the information obtained will be expanded. In the case of the additional information requested by the reviewer regarding wood debarking, its genesis and its categorization, as well as its meaning and importance, there is abundant literature on this topic; for instance the article by Genova et al. (2015) and other collections recently published (Díez-Herrero et al, 2013; Benito & Díez-Herrero, 2015; Ballesteros-Canovas et al., 2015.). These works provide detailed explanations of search procedures, classification, interpretation and the quality of data derived from dendrogeomorphologic evidence. Nevertheless, if the reviewer and the editor believe that including more information on the above is appropriate, we are willing to add more detail to the manuscript regarding these methodological aspects.

2- A vital result of the paper is the large discrepancy in the model runs resulting in 50 m3/s and 1200 m3/s. There is a very extensive discussion on uncertainties in data and methods, which is honorable. From my point of view the discrepancy of the results is that large so talking about "uncertainties" is critical. Isn't it more a signal, that either data or methods are simply not suitable to answer the research question? Where is the border between uncertainty and infeasibility? Especially since both precipitation and dendrogeomorphological data include major uncertainties the conclusion that the

obtained results could improve flood hazard and risk analysis is questionable.

Undoubtedly, conducting an analysis such as the one performed here can entail a significant degree of uncertainty. However, from the authors' point of view, the combination of data sources used reduces this uncertainty by attempting to relate indirect flash flood evidence to the theoretical clear water flood which could have caused them. The absence of flow data derived from flow gauges in the basin does severely limit calibration of any type of hydrological model. We are, therefore, aware of the possible sources of uncertainty regarding the results of the analysis, although the possibility of reducing these uncertainties is minimal. Consequently, doubts about the efficiency or applicability of studies such as the one presented here can be raised. However, from the authors' point of view, if these limitations invalidate the studies conducted in such basins we will be giving up a chance to investigate and attempt to understand how these small ungauged mountain basins work. This type of basins represents a very high percentage of the total, especially in developing countries, and from the point of view of flood risk and land use management, this lack of knowledge does not seem to be a good idea. Moreover, from the point of view of scientific research, real advances and innovations occur when researchers work on solving problems in basins with missing data and high uncertainties, but not when known methods are applied in basins with abundant and well known data, which would be mere repetitions of technical reports.

3- Minor comments: P5, L6: Include more information on dendrogeomorphological data and methods (see above).

As we mentioned above, a detailed description of the dendrogeomorphological data used in this work can be found in Genova et al., 2015, and we have thus omitted it. However, if the reviewers consider it necessary, we will include this information in the revised version and also mention some other papers such as Díez-Herrero et al. (2013); Benito and Díez-Herrero (2015); and Ballesteros-Cánovas et al. (2015).

4- Minor comments: P6, L10: The POT extreme value statistics is based on data lasting

<24 hours and >24 hours. Please explain this more in detail. From my point of view one has to choose defined duration levels (e.g. 6 hours, 12 hours etc.) and perform the statics for each series individually.

In this case it is possible that the authors have not clearly explained the work done. Statistical analysis to obtain quantiles of precipitation using the GP-POT distribution function was conducted on the daily (24 h) rainfall series. We will correct the wording to make this clearer.

5- Minor comments: P7, L24: Please explain more in detail how the v1997 topography was built.

To generate the v1997 topography, we combined two topographic surfaces: the current topography (v2009) and the 3D surface that adjusts the topographic position of the base of the trees when dendrogeomorphological information is available. The criterion used to combine both topographical surfaces was to conserve the data conservation with the highest elevation. Based on that criterion, the resulting combined surface retained the current morphology of the slopes outside the channel while, in the streambed and banks, the surface was obtained from the adjustment of the location of the dendrogeomorphological data.

6- Minor comments: P10, L11: The explanations given in sec. 4.3 can only hardly justify this large discrepancy found in the model results.

The authors believe that the combination of several of the factors stated in section 4.3 of the article can justify the discrepancies shown by the models. In the authors' opinion, the differences in precipitation between the weather station and the entire basin itself need to be considered because this would indicate that the precipitations could have been significantly greater, and perhaps more intense. It should also be considered, in the authors' view, that because of the characteristics of the basin and its high contribution of sediment and floating solid load (woody material) to the flood, the flow characteristics can vary significantly compared to a flood that is only composed of

liquid flow. In this sense, with a smaller volume of liquid flow (which therefore requires a smaller volume of precipitation), the total volume (solid + liquid) of the avenue will be greater. Increasing the flow associated with the model that only considers precipitation and reducing the liquid volume of the flow associated with the model that takes into account the dendrogeomorphological data would reduce differences in the flow rates obtained by the two models considered.

7- Minor comments: Figure 8: Please add the original data (plotting positions) in the plot; the differences in the results seems a little bit strange. Please check also the whole calculation.

As suggested by the anonymous reviewer # 1, the statistical analysis performed and shown in the article will be reviewed in its entirety, to prevent any errors or inconsistencies in the present analysis.